# Differential Effects of Whole Red Raspberry Polyphenols and Their Gut Metabolite Urolithin A on Neuroinflammation in BV-2 Microglia

**DOI:** 10.3390/ijerph18010068

**Published:** 2020-12-24

**Authors:** Ashley Mulcahy Toney, Mahaa Albusharif, Duncan Works, Luke Polenz, Stacie Schlange, Virginia Chaidez, Amanda E. Ramer-Tait, Soonkyu Chung

**Affiliations:** 1Department of Nutrition and Health Sciences, University of Nebraska-Lincoln, Lincoln, NE 68583, USA; amulcahy@huskers.unl.edu (A.M.T.); duncan.works@huskers.unl.edu (D.W.); Luke.polenz@unmc.edu (L.P.); stacie.schlange@unmc.edu (S.S.); vchaidez2@unl.edu (V.C.); 2Department of Food Science and Technology, University of Nebraska-Lincoln, Lincoln, NE 68588, USA; malbusharif@huskers.unl.edu (M.A.); aramer-tait2@unl.edu (A.E.R.-T.); 3Nebraska Food for Health Center, University of Nebraska-Lincoln, Lincoln, NE 68588, USA; 4Department of Nutrition, University of Massachusetts-Amherst, Amherst, MA 01003, USA

**Keywords:** red raspberry polyphenols, urolithin A, ellagic acid, neuroinflammation, microglia, inflammation, JNK

## Abstract

Whole red raspberry polyphenols (RRW), including ellagic acid, and their gut-derived metabolite, urolithin A (UroA), attenuate inflammation and confer health benefits. Although results from recent studies indicate that polyphenols and UroA also provide neuroprotective effects, these compounds differ in their bioavailability and may, therefore, have unique effects on limiting neuroinflammation. Accordingly, we aimed to compare the neuroprotective effects of RRW and UroA on BV-2 microglia under both 3 h and 12 and 24 h inflammatory conditions. In inflammation induced by lipopolysaccharide (LPS) and ATP stimulation after 3 h, RRW and UroA suppressed pro-inflammatory cytokine gene expression and regulated the JNK/c-Jun signaling pathway. UroA also reduced inducible nitric oxide synthase gene expression and promoted M2 microglial polarization. During inflammatory conditions induced by either 12 or 24 h stimulation with LPS, UroA—but not RRW—dampened pro-inflammatory cytokine gene expression and suppressed JNK/c-Jun signaling. Taken together, these results demonstrate that RRW and its gut-derived metabolite UroA differentially regulate neuroprotective responses in microglia during 3 h versus 12 and 24 h inflammatory conditions.

## 1. Introduction

The incidence of neurological disorders and diseases, such as Alzheimer’s disease, is increasing worldwide [1]. Neuroinflammation drives the development of these diseases as resident macrophage-like cells, microglia, become chronically activated and induce a pro-inflammatory environment in the brain [2]. Microglia survey the surrounding area for signs of damage to nearby cells or tissues [3]. Under homeostatic conditions, microglia are alternatively activated and promote neuroprotection by adopting an M2 polarization state. However, when microglia detect damage-associated molecular patterns (DAMPs), they become classically activated, polarize to an M1 phenotype, and induce a pro-inflammatory environment [3]. Because prolonged oxidative stress and inflammation ultimately result in the development of neurodegenerative diseases, interventions targeting classical microglial activation are needed.

Red raspberries have been studied extensively for their anti-inflammatory properties and their ability to attenuate inflammation and oxidative stress in multiple chronic diseases, including cardiovascular disease, obesity, cancer, and Alzheimer’s disease [4,5]. The United States is the third largest red raspberry producer, with an industry valued at more than $270 million according to estimates in 2014, suggesting consumption of raspberries has a positive impact on berry growers [6]. Moreover, the United States exports an estimated 57.6 million pounds of raspberries to other countries but also imports raspberries worldwide to meet consumer demands [6]. Raspberries are rich in polyphenols, including anthocyanins and ellagitannins. Ellagic acid (EA), a free form of ellagitannin (ET), resists hydrolysis in the stomach and subsequently undergoes microbial conversion into urolithins in the gut [5]. Of these gut-derived metabolites, urolithin A (UroA) has been studied extensively and can achieve bioavailability in human plasma once conjugated by phase II biotransformation in the liver [7,8]. Although most polyphenolic compounds have low bioavailability, increased concentrations of UroA can be detected in the plasma and brains of rats fed pomegranates rich in ellagic acid, thereby suggesting that UroA can provide neuroprotective effects in vivo by passing through the blood brain barrier [9,10,11]. Moreover, free UroA has been detected in circulation and peripheral tissues during inflammatory conditions [12].

Numerous studies have specifically documented the ability of red raspberry polyphenols or their gut-derived metabolites, urolithins, to attenuate inflammation and oxidative stress [13,14,15,16]. EA, as well as its digested metabolites, have been shown in vitro to protect neuroblastoma cells against oxidative stress [10]. UroA, in particular, is known to decrease neuroinflammation in both in vitro and in vivo models via regulation of autophagy and SIRT-1 activation [17] and MAPK/Akt/NF-ĸB signaling pathways [18]. Although studies indicate that c-Jun N-terminal kinase (JNK) signaling promotes pro-inflammatory responses in microglia [19,20], its role in UroA-mediated neuroprotection has not been assessed. Despite these promising results, concerns regarding the bioaccesibility of raspberry polyphenols warrants a direct comparison of the beneficial effects of polyphenols and their gut metabolite, UroA, in regulating cytokine responses and oxidative stress signaling pathways during both 3 h and 12 and 24 h neuroinflammatory conditions.

The aim of this study, therefore, was to compare the neuroprotective effects of red raspberry polyphenols from whole fruit (RRW) versus UroA in BV-2 microglia cells using two different time windows (3 h versus 12 and 24 h induced inflammation). We report that UroA and RRW exhibited differential temporal regulation of pro-inflammatory gene expression and JNK/c-Jun signaling pathways in BV-2 microglia.

## 2. Materials and Methods

### 2.1. Whole Red Raspberry Polyphenol Preparation

Freshly frozen “Wakefield” red raspberries were donated by Enfield Farms (Lynden, WA) and were used for this study. Extraction of red raspberry polyphenols from whole fruits (i.e., RRW) was conducted using a protocol previously described by our laboratory [21]. Phenolic composition of RRW extracts is found in Appendix A. Stock concentrations of RRW (20 mg/mL dissolved in PBS) were stored at −80 °C and diluted fresh immediately prior to the start of each experiment.

### 2.2. Cell Culture

Murine BV-2 microglia were provided by Dr. Janos Zempleni from the University of Nebraska, Lincoln. Urolithin A (UroA) was obtained as previously described [22]. UroA stock concentrations (10 mM) were prepared in DMSO and freshly diluted using PBS (HyClone, Marlborough, MA, USA). For all experiments, cells were cultured in High Glucose Dulbecco’s Minimal Essential Media containing 4.0 mM L-glutamine without sodium pyruvate (Gibco; Thermo Fisher Scientific, Waltham, MA, USA) supplemented with 10% fetal bovine serum (Atlanta Biologicals, Flowery Branch, GA, USA), penicillin (100 IU/mL), and streptomycin (100 µg/mL) (HyClone). Cells were maintained in a humidified atmosphere at 37 °C and 5% CO_2_. Medium was changed every 36–48 h. BV-2 cells were seeded at 4 × 10^5^ cells per 6-well plate for all experiments.

To investigate the role of RRW and UroA in microglial inflammation, BV-2 cells were pretreated with either 10 µg/mL RRW or 10 µM UroA for 12 h. Cells were then subjected to either lipopolysaccharide (LPS, 500 ng/mL) treatment for 3 h followed by ATP (2.5 mM) stimulation for 30 min in Opti-MEM (Gibco) or LPS (500 ng/mL) treatment for 12 or 24 h. Concentrations of RRW polyphenols (10 μg/mL) and UroA (10 μM) were selected because they have been previously shown to induce beneficial responses in BV-2 microglia [17] as well as J774 and bone marrow-derived macrophages (BMDM) [21]. To investigate the effects of RRW and UroA on M2 polarization, BV-2 cells were pre-treated with either RRW (10 μg/mL) or UroA (10 μM) for 12 h and then stimulated with interleukin (IL)-4 and IL-13 (Peprotech, Cranbury, NJ, USA) in Opti-MEM for 24 h.

### 2.3. Total RNA Extraction and RT-qPCR

Total RNA was isolated using the TRIzol Reagent (Invitrogen, Carlsbad, CA, USA). RNA concentrations were measured using a Synergy HT Microplate Reader (BioTek, Winooski, VT, USA; Gen5 Version 3.08), and 2 μg of mRNA was converted into cDNA in a total volume of 20 μL (iScript cDNA Synthesis Kit, Bio-Rad, Hercules, CA, USA). Gene-specific primers for qRT-PCR (Appendix A) were purchased from Integrated DNA Technologies (Coralville, IA, USA). Gene expression was determined by real-time qPCR (Quant Studio 5, Applied Biosystems, ThermoFisher Scientific, (Waltham, MA, USA) using SYBR green (Thermo Scientific, Waltham, MA, USA). Relative gene expression was determined based on the 2^-ΔΔCT^ method with normalization of the raw data to hypoxanthine-guanine phosphoribosyltransferase (*Hprt*).

### 2.4. Western Blot Analysis

Cell extracts were prepared in RIPA buffer containing protease (Millipore Sigma, Burlington, MA, USA) and phosphatase inhibitors (Millipore Sigma) (2 mM Na_3_VO_4_, 20 mM β-glycerophosphate, and 10 mM NaF). Total protein concentrations were determined using the Pierce BCA Protein Assay Kit (ThermoFisher, Waltham, MA, USA). Cell culture supernatants were collected and centrifuged (10,000× *g*, 5 min) to remove cell debris. Supernatant protein was precipitated using the trichloroacetic acid (TCA) precipitation method. Briefly, one volume of 100% (*w*/*v*) TCA stock was added to four volumes of supernatant sample, incubated overnight at 4 °C, and centrifuged at 15,000× *g* for 10 min. Supernatants were removed and the pellet was washed with 200 μL cold acetone at 15,000× *g* for 10 min. Pellets were dried in a 95 °C heat block for 5 min to evaporate remaining acetone and resolved with water and SDS sample loading buffer (LI-COR Biosciences, Lincoln, NE, USA).

Proteins (10 μg for cell extracts and total precipitated cell supernatants) were fractionated using 8–12% sodium dodecyl sulfate (SDS) polyacrylamide gel electrophoresis (PAGE) and transferred to polyvinylidene fluoride low-fluorescent (PVDF) membranes (Millipore Sigma). Membranes were blocked (Odyssey blocking buffer, LI-COR) for 1 h at room temperature (RT), washed with 1 × TBS-T (20 mM Tris-HCl + 150 mM NaCL + 0.01% Tween 20), and incubated with primary antibody (Appendix A) in blocking buffer at 4 °C overnight and then the corresponding secondary antibody (Appendix A) in blocking buffer containing 0.05% Tween-20 and 0.01% SDS at RT for 1 h. Blots were scanned using the Odyssey Infrared Imaging System (LI-COR, Lincoln, NE, USA) and analyzed with Image Studio Lite Software Ver 5.2.5 (LI-COR, Lincoln, NE, USA). Total protein for each blot was measured using the Revert^TM^ 700 Total Protein Stain Kit (LI-COR, Lincoln, NE, USA) and used for normalization and quantification.

### 2.5. Statistical Analysis

All data were analyzed using one-way ANOVA followed by Tukey’s multiple comparison tests * *p <* 0.05 or ** *p <* 0.01. All analyses were preformed using Graph Pad Prism (Ver 8.3.1., La Jolla, CA, USA).

## 3. Results

### 3.1. Whole Red Raspberry Polyphenols and Urolithin A Decreased Neuroinflammatory Cytokine Expression

Previous research has reported that LPS and ATP induce an inflammasome response in this cell line [23]. Although these conditions increased expression of *Nlrp3*, they did not induce changes in the expression of other inflammasome construct genes (e.g., *Asc* and *Tlr4*), nor did they yield detectable levels of activated caspase-1 in culture supernatants (Appendix A). However, LPS and ATP stimulation significantly upregulated expression of genes encoding for the pro-inflammatory cytokines interleukin-1 beta (*Il1b*), interleukin-6 (*Il6*), and tumor necrosis factor alpha (*Tnf*) compared to unstimulated cells (Figure 1). Both RRW and UroA significantly decreased expression of *Il1b* and *Il6* compared to untreated cells stimulated with LPS and ATP (Figure 1). UroA, but not RRW, significantly decreased *Tnf* gene expression after LPS and ATP treatment (Figure 1). Taken together, these results suggest that UroA, the gut-derived metabolite from RRW, and, in part, RRW can protect against neuroinflammation induced by LPS and ATP by limiting pro-inflammatory cytokine responses.

### 3.2. Whole Red Raspberry Polyphenols and Urolithin a Regulated JNK/c-Jun Signaling Pathways during LPS and ATP-Induced Inflammation

Previous literature has reported that UroA regulates the mitogen-activated protein kinase (MAPK)/NF-κB pathways to protect against neuroinflammation [18,24], but the role of the JNK/c-Jun pathway in UroA-mediated protection remains unknown. We therefore investigated the effect of RRW and UroA on c-Jun N-terminal kinases (JNK) and subsequent p-c-Jun protein production during LPS and ATP-induced neuroinflammation. Both p-JNK and p-c-Jun protein expression was increased following treatment with LPSplus ATP compared to treatment with only LPS (Figure 2), an observation consistent with previous reports connecting the c-Jun pathway to stress and inflammation in microglia [20]. Importantly, RRW and UroA significantly decreased the phosphorylation of both p-JNK and p-c-Jun compared to untreated microglia stimulated with LPS and ATP (Figure 2). These results, therefore, suggest that regulation of the JNK/c-Jun pathways may contribute to neuroprotective effects by polyphenols and polyphenol-derived metabolites by limiting microglial activation and stress.

### 3.3. Urolithin A Reduced iNOS Gene Expression and Promoted M2 Microglia Polarization

During 3 h inflammatory responses, low levels of inducible nitric oxide synthase (iNOS) can protect the central nervous system from injury; however, chronically activated microglia generate high levels of iNOS, which is detrimental and indicative of neuroinflammation [25]. We observed a significant increase in *iNos* expression after stimulating BV-2 cells with LPS and ATP (Figure 3A). However, both RRW and UroA significantly decreased *iNos* expression (Figure 3A).

Previously, we demonstrated that UroA induces M2 macrophage polarization in BMDM [26]. We therefore tested whether UroA could similarly induce M2 polarization in microglia. We observed a significant increase in the expression of the Chitinase-like 3 gene (*Ym1*; also known as *Chi313*) in BV-2 microglia pretreated with UroA compared to control upon M2 stimulation with IL-4 plus IL-13 alone (Figure 3B). Notably, RRW treatment did not alter *Ym1* expression. Collectively, these results indicate that UroA promotes M2 polarization in microglia similar to other macrophage-like cells, which may be a unique mechanism by which UroA, but not red raspberry polyphenols, mediates neuroprotective effects.

### 3.4. Urolithin A Decreased Neuroinflammation in a 12 and 24 h LPS Treatment Model

We further investigated the effect of RRW and UroA on attenuating inflammation using a longer exposure to LPS for 12 or 24 h. After 12 h of LPS treatment, we observed a significant increase in the expression of *Il1b*, *Il6*, and *Tnf* compared to untreated microglia (Figure 4A). Treatment with LPS with RRW significantly increased *Il1b* and *Il6* expression but decreased *Tnf* expression compared to LPS alone. Notably, treatment with LPS with UroA significantly decreased *Il1b, Il6,* and *Tnf* (Figure 4A). We further assessed the effects of RRW and UroA on pro-inflammatory cytokine gene expression in microglia following a longer exposure to LPS for 24 h. Similar to a 12 h LPS stimulation, we observed significant increases in *Il1b*, *Il6*, and *Tnf* after LPS treatment for 24 h, which were all attenuated with UroA treatment (Figure 4B). However, unlike the patterns observed after the 12 h treatment, RRW decreased expression of *Il6* and *Tnf* and had no significant effect on *Il1b* expression compared to microglia treated with LPS alone for 24 h (Figure 4B). Taken together, these results suggest UroA, but not RRW, exerts neuroprotective effects during 12 and 24 h LPS exposure.

### 3.5. Urolithin A Regulated the JNK Signaling Pathway upon 12 and 24 h LPS Treatment

Because UroA uniquely limited pro-inflammatory cytokine gene expression in microglia, we subsequently investigated the ability of UroA to downregulate the JNK/c-Jun pathways during 12 and 24 h LPS exposure. At both timepoints, LPS treatment significantly increased phosphorylation of JNK compared to untreated cells but had no effect on the p-c-Jun pathway (Figure 5). After 12 h, UroA decreased p-JNK protein expression in microglia compared to treatment with LPS alone (Figure 5A). Consistent with proinflammatory gene expression, RRW treatment had no effect on p-JNK under these conditions (Figure 5A).

Similar to the 12 h LPS exposure, UroA, but not RRW, down-regulated the JNK pathway in microglia after 24 h treatment of LPS (Figure 5B). Of note, we observed ~67% less p-JNK protein expression at 24 h compared to 12 h for all treatments (Figure 5B versus 4B). In total, these results demonstrate that UroA, but not RRW polyphenols, downregulate the JNK signaling pathway in response to 12 and 24 h LPS treatment.

## 4. Discussion

The current study was designed to investigate whether whole red raspberry polyphenols (RRW) or UroA, the gut-derived metabolite from EA, could attenuate neuroinflammation in BV-2 microglia. We found that both UroA and RRW decreased pro-inflammatory cytokine expression and JNK/c-Jun pathway activation in response to the 3 h (LPSplus ATP) induced inflammation. UroA also limited pro-inflammatory responses during the 12 and 24 h induced neuroinflammation by LPS; however, RRW had neutral or only minor effects. Moreover, we observed that both UroA and RRW reduced *iNos* levels, but only UroA promoted *Ym1* expression for neuroprotection. Together, these results indicate a differential temporal regulation of neuroinflammation for RRW and UroA.

By using a whole food approach in the present study, we are able to directly compare the bioactivity of whole red raspberry polyphenols to that of their selective gut metabolite UroA against neuroinflammation. In recent years, interest in the ability of UroA to provide neuroprotection has increased considerably because this metabolite crosses the blood brain barrier [11]. Additionally, oral administration of the synthetic gut-metabolite, UroA, or its precursor EA, has been shown to be neuroprotective in mice [27]. However, determining whether polyphenols from red raspberries, which consist of ellagitannins as well as anthocyanins, confer this same level of protection as the gut metabolites remains unclear.

Our study uniquely evaluated a polyphenolic extract from whole red raspberries that was stripped of any sugars and fibers capable of conferring confounding biological activities. Moreover, our extract contained all of the major polyphenols found in whole red raspberries, including quercetin, myricetin, EA, and anthocyanins, which enabled us to test the neuroprotective effects of whole fruit polyphenols rather than focusing on a concentrated form of EA, as has been studied previously [21]. Although RRW provided robust protection against neuroinflammation in LPS/ATP-stimulated microglia, RRW was only partly effective in reducing pro-inflammatory cytokine expression during 12 and 24 h LPS exposure, and it had no significant impact on JNK signaling. This result stands in contrast to observations made using a N9 murine microglial model, where a post in vitro digestion model of raspberries rich in ellagitannin and EA provided neuroprotection for up to 24 h post-LPS stimulation by suppressing the MAPK/NFAT/NF-ĸB signaling pathways [10]. In another study, EA pre-treatment decreased TNF-ĸ and NO production from BV-2 microglia by attenuating MyD88/NF-ĸB and p38/Erk/JNK signaling pathways after stimulation with a high dose of LPS (1 ug/mL) for 24 h [28,29]. The inconsistent potency of our whole red raspberry polyphenol extract with respect to regulating neuroinflammatory responses may stem from its composition, as it contains not only free EA but also significant amounts of anthocyanins and epicatechins. Further studies evaluating the contribution of these additional polyphenols to neuroprotection are clearly warranted.

Although whole fruit polyphenols provided limited benefits to microglia, we observed a strong and robust neuroprotective effect for UroA, the gut-derived metabolite of RRW, during both 3 h and 12 and 24 h inflammatory conditions. Consistent with our results, others have also shown that UroA attenuates inflammation in BV-2 microglia stimulated with LPS for 24 h [17] and regulates pro-inflammatory cytokine expression and MAPK signaling pathways, including the ERK/p38 signaling pathway [18]. However, the effects of UroA and RRW following induction of acute neuroinflammation with ATP and LPS representative of ischemia [30,31] have not been investigated. Extracellular ATP is a DAMP leaked from injured brain cells that promotes neuroinflammation. ATP mediates inflammasome activation and oxidative stress in microglia by acting on purinergic receptors to promote expression of pro-inflammatory cytokines in addition to activating the P2X7 receptor and downstream JNK signaling pathways [32]. ATP is also able to activate MAPK signaling pathways in BV-2 microglia [33]. We observed that both RRW and UroA suppressed 3 h LPS+ATP-induced inflammation driven by LPS plus ATP, possibly via antioxidant activities. Further studies are encouraged to understand the mechanisms underlying why RRW and its gut-derived metabolite, UroA, decrease inflammation acutely.

M2 microglia are thought to provide neuroprotective effects following neuroinflammation or associated damage. However, under chronic inflammatory conditions, microglia may be unable to maintain an M2 polarized state [3]. Previously, our laboratory showed that UroA was able to promote M2 macrophage polarization in BMDM due to its ability to promote mitochondrial biogenesis [26]. Several studies suggest M2 microglial polarization is associated with mitochondrial flux rather than glycolytic flux, the latter of which is prominent in BV-2 and primary murine microglia stimulated with LPS [34]. Other work has shown that mitochondrial uncoupling protein-2 (UCP2) promotes M2 microglial polarization and prevents neuroinflammation [35]. Moreover, activation of M2 polarization has been shown to suppress microglial pro-inflammatory cytokine production and provide neuroprotection [36], which is consistent with our findings. Collectively, these reports support the concept that UroA may promote M2 microglial polarization via regulation of mitochondrial flux.

As neurodegenerative diseases continue to rise, investigating potential therapeutics that target pathways of neuroinflammation are imperative. Our findings and those of others support the potential for UroA to provide neuroprotective effects. Some studies have found that the gut microbiome may be crucial in converting EA-rich foods into bioactive urolithins [37,38,39], while others have evaluated direct oral administration of UroA to healthy human adults [40]. However, it remains to be investigated whether these approaches provide neuroprotective effects to populations at risk of developing neurodegenerative diseases. Mechanistically, UroA and, to a lesser extent, RRW may provide neuroprotection by regulating the JNK signaling pathway. In ischemia, microglial activation upregulates JNK signaling pathways and subsequently increases the secretion of pro-inflammatory cytokines [41]. By suppressing JNK and NF-ĸB downstream signaling with a JNK-In-8 JNK inhibitor, researchers have found that neuroinflammation is inhibited and neurological function is improved post-ischemic brain injury [41]. Perhaps nanotechnology-based delivery vehicles aimed at improving the bioavailability of EA and/or urolithins [42] can, one day, be used to target JNK/NF-ĸB signaling and provide neuroprotective benefits.

## 5. Conclusions

Our study demonstrated the differential impact of red raspberry polyphenols and their gut-derived metabolite of UroA on neuroprotection in BV-2 microglia. UroA was able to attenuate pro-inflammatory cytokine gene expression during both 3 h and 12 and 24 h LPS-induced neuroinflammation, in part, by suppressing the JNK signaling pathway and possibly promoting M2 macrophage polarization. Moreover, RRW suppressed neuroinflammation following 3 h LPS/ATP stimulation, but lost its ability to provide neuroprotection in a 12 and 24 h induced inflammatory setting.

## Figures and Tables

**Figure 1 ijerph-18-00068-f001:**
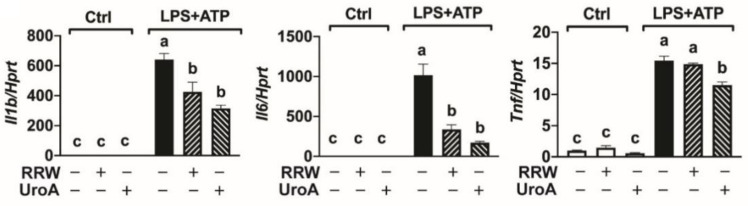
Both whole red raspberry polyphenols and urolithin A decreased pro-inflammatory cytokine gene expression. V-2 cells treated with LPS (500 ng/mL) for 3 h followed by ATP (2.5 mM) for 30 min with or without whole red raspberry polyphenols (10 µg/mL) or urolithin A (10 µM). Gene expression levels measured by RT-qPCR for *Il1b, Il6*, and *Tnf.* All values are represented as the mean ± SEM. Treatments with different letters indicate significant differences by one-way ANOVA, *p* < 0.05. LPS = lipopolysaccharide; RRW = whole red raspberry polyphenols; UroA = urolithin A.

**Figure 2 ijerph-18-00068-f002:**
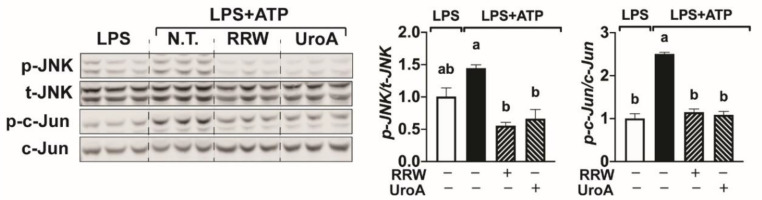
Both whole red raspberry polyphenols and urolithin A attenuated neuroinflammation and regulated JNK/c-Jun signaling pathways. BV-2 cells treated with LPS (500 ng/mL) for 3 h followed by ATP (2.5 mM) for 30 min with or without whole red raspberry polyphenols (10 µg/mL) or urolithin A (10 µM). Protein expression from cell lysates for p-JNK, t-JNK, p-c-Jun, and c-Jun by Western blot analysis. Relative protein intensities normalized to total-JNK or c-Jun by LICOR Image Studio Lite (right). All values are represented as the mean ± SEM. Treatments with different letters indicate significant differences by one-way ANOVA, *p* < 0.05. LPS = lipopolysaccharide; RRW = whole red raspberry polyphenols; UroA = urolithin A.

**Figure 3 ijerph-18-00068-f003:**
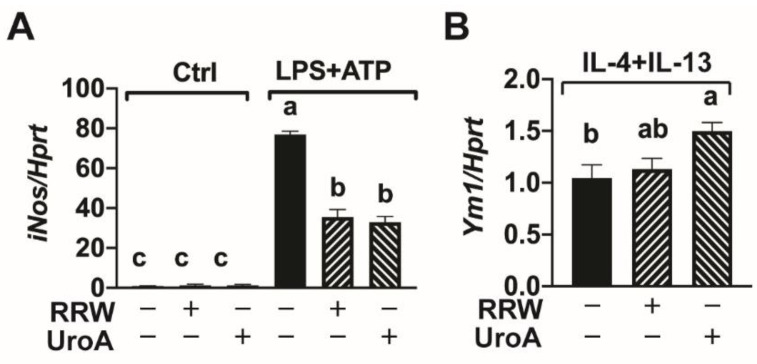
Urolithin A attenuated inducible nitric oxide synthase expression and promoted M2 microglial polarization. (**A**) BV-2 cells treated with LPS (500 ng/mL) for 3 h followed by ATP (2.5 mM) for 30 min with or without whole red raspberry polyphenols (10 µg/mL) or urolithin A (10 µM). Gene expression levels measured by RT-qPCR for *iNos*. (**B**) BV-2 cells treated with IL-4 (100 ng/mL) and IL-13 (10 ng/mL) for 24 h with or without whole red raspberry polyphenols (10 µg/mL) or urolithin A (10 µM). Gene expression levels measured by RT-qPCR for *Ym1*. All values are represented as the mean ± SEM. Treatments with different letters indicate significant differences by one-way ANOVA, *p* < 0.05. IL = interleukin; LPS = lipopolysaccharide; RRW = whole red raspberry polyphenols; UroA = urolithin A.

**Figure 4 ijerph-18-00068-f004:**
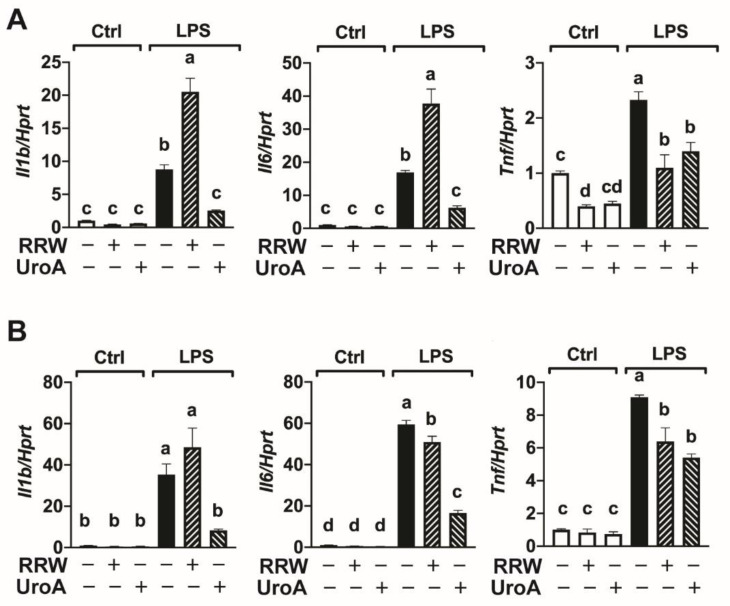
Urolithin A, but not whole red raspberry polyphenols, attenuated pro-inflammatory cytokine gene expression upon 12 and 24 h LPS treatment. (**A**) BV-2 cells treated with LPS (500 ng/mL) for 12 h with or without whole red raspberry polyphenols (10 µg/mL) or urolithin A (10 µM). Gene expression for *Il1b, Il6*, and *Tnf* by RT-qPCR. (**B**) BV-2 cells treated with LPS (500 ng/mL) for 24 h with or without whole red raspberry polyphenols (10 µg/mL) or urolithin A (10 µM). Gene expression for *Il1b, Il6*, and *Tnf* by RT-qPCR. All values represented as the mean ± SEM. Treatments with different letters indicate significant differences by one-way ANOVA, *p* < 0.05. LPS = lipopolysaccharide; RRW = whole red raspberry polyphenols; UroA = urolithin A.

**Figure 5 ijerph-18-00068-f005:**
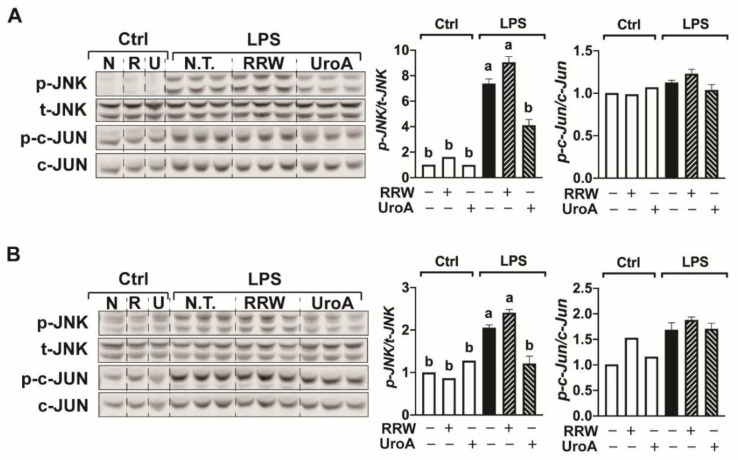
Urolithin A, but not whole red raspberry polyphenols, regulated the JNK signaling pathway upon 12 and 24 h LPS treatment. (**A**) BV-2 cells treated with LPS (500 ng/mL) for 12 h with or without whole red raspberry polyphenols (10 µg/mL) and urolithin A (10 µM). Protein expression from cell lysates for p-JNK, t-JNK, p-c-Jun, c-Jun by Western blot analysis. (**B**) BV-2 cells treated with LPS (500 ng/mL) for 24 h with or without whole red raspberry polyphenols (10 µg/mL) and urolithin A (10 µM). Protein expression from cell lysates for p-JNK, t-JNK, p-c-Jun, and c-Jun by Western blot analysis. Relative protein intensities normalized to total-JNK or c-Jun by LICOR Image Studio Lite (right). All values represented as the mean ± SEM. Treatments with different letters indicate significant differences by one-way ANOVA, *p* < 0.05. Ctrl = control; N and N.T. = no treatment; W and RRW = whole red raspberry polyphenols; U and UroA = urolithin A.

## Data Availability

All data is available based on “MDPI Research Data Policies” at https://www.mdpi.com/ethics.

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
