# Peer review of "Differential Effects of Whole Red Raspberry Polyphenols and Their Gut Metabolite Urolithin A on Neuroinflammation in BV-2 Microglia"

_ijerph, 2020, doi:10.3390/ijerph18010068_

Round 1

Reviewer 1 Report

Toney et al gave us a very intersting report and the study is meaningful for agricultural industry especially for red raspberry development and functional foods production. After clarify some issues, the manuscript could be acceptable for publication.

  1. The outputs of red raspberry in US and world should be provided.

    2. The kinds of polyphenols in red raspberry should be clarified using HPLC or UPLC-QTOF technology.

    3. It's better to explain how to make sure the same contents of whole polyphenols each batch.

    4. In the method of western-blotting, 10 g protein seems too much for SDS-PAGE and transfer membranes. It should be checked again.

Author Response

Reviewer 1

Comments and Suggestions for Authors

Toney et al gave us a very interesting report and the study is meaningful for agricultural industry especially for red raspberry development and functional foods production. After clarify some issues, the manuscript could be acceptable for publication.

We thank the reviewer for their positive comments on our work.

  1. The outputs of red raspberry in US and world should be provided.

We thank the reviewer for their suggestion about connecting red raspberries and its relevance for production in the United States and world at large. We included this information in the introduction (lines 44-48).

  1. The kinds of polyphenols in red raspberry should be clarified using HPLC or UPLC-QTOF technology.

Thank you for your suggestion in providing information regarding the types of polyphenols found in our red raspberry extract. Previously, we published this data in Fan et al., 2020. We have updated the manuscript and included this table as Supplementary Table 1 and renamed Table S1 with Primer Sequences as Table S2 and the previous Table S2 as Table S3 for the list of antibodies used in western blotting. These changes have been updated in the supplementary file on page 1.

  1. It's better to explain how to make sure the same contents of whole polyphenols each batch.

Thank you for the opportunity to clarify how controlled for the contents of whole polyphenols in each batch. After ion exchange chromatography and lyophilization using 3kg of fresh-frozen whole red raspberries, the red raspberry polyphenol extract in powder form was weighed and suspended in 1x PBS. Afterwards, the stock was vortexed and aliquoted into various tubes to prevent freeze/thaw cycles. We used the same stock which can be found in our previous manuscript (Fan et al., 2020).

  1. In the method of western-blotting, 10 g protein seems too much for SDS-PAGE and transfer membranes. It should be checked again.

It was a typo. We updated the manuscript to reflect 10 mg – not 10 g of protein for SDS-PAGE (line 113).

Reviewer 2 Report

This paper first reveals that whole red raspberry polyphenols and  its gut-derived metabolite urolithin A differentially regulate neuroprotective responses, which JNK/c-Jun signaling pathway and decreased neuroinflammatory cytokine expression in microglia during acute versus chronic inflammatory conditions. These results are of great theoretical significance and practical value for attenuating inflammation and health benefits. The analysis process is comprehensive, good organized, advanced research technology, large amount of information and so on. Minor revision can be published in International Journal of Environmental Research and Public Health. However, there are some major issues need to be improved:

  1. Introduction: Authors should supplement the distribution, annual output, utilization value of  global red raspberry, and expand the readers of the paper.
  2. Materials and Methods: The first letter of the second word is in lowercase and the full text should be uniform from 2.1 to 2.5.
  3. Results: Authors should add highlighting the molecular mechanisms of neuroprotectionfor polyphenols and urolithin A. 
  4. Discussion: Authors should increase thathealth effects and molecular mechanisms of polyphenols to enhance the reader's interest, such as references: https://www.mdpi.com/1422-0067/19/9/2785;https://www.hindawi.com/journals/omcl/2020/3836172/
  5. References:  Authors shouldrevise the format of reference according to IJERPH Journal.

Author Response

Reviewer 2

 Comments and Suggestions for Authors

This paper first reveals that whole red raspberry polyphenols and its gut-derived metabolite urolithin A differentially regulate neuroprotective responses, which JNK/c-Jun signaling pathway and decreased neuroinflammatory cytokine expression in microglia during acute versus chronic inflammatory conditions. These results are of great theoretical significance and practical value for attenuating inflammation and health benefits. The analysis process is comprehensive, good organized, advanced research technology, large amount of information and so on. Minor revision can be published in International Journal of Environmental Research and Public Health. However, there are some major issues need to be improved:

  1. Introduction: Authors should supplement the distribution, annual output, utilization value of global red raspberry, and expand the readers of the paper.

This comment was addressed under Reviewer 1. We thank the reviewer for their suggestion about connecting red raspberries and its relevance for production in the United States and world at large. We included this information in the introduction (lines 44-48).

  1. Materials and Methods: The first letter of the second word is in lowercase and the full text should be uniform from 2.1 to 2.5.

We thank the reviewer for bringing this to our attention. We have made these updates accordingly in 2.1 to 2.5.

  1. Results: Authors should add highlighting the molecular mechanisms of neuroprotection for polyphenols and urolithin A. 

Thank you for the suggestion in highlighting the molecular mechanisms conferring neuroprotection for red raspberry polyphenols and urolithin A. Currently, our study provides preliminary data regarding downstream molecular pathways to JNK/C-Jun signaling. However, we provided our interpretation of the mechanism within the discussion. We infer the red raspberry polyphenols are providing protection via antioxidant activities; however, urolithin A provides robust protection through JNK/c-Jun signaling and other pathways posited from other studies such as MAPK/NFAT/NFkB signaling pathways (lines 275-278, 293-294). However, more studies are needed to definitively conclude the molecular mechanism that confers these neuroprotective effects by polyphenols and urolithin A.

  1. Discussion: Authors should increase that health effects and molecular mechanisms of polyphenols to enhance the reader's interest, such as references: https://www.mdpi.com/1422-0067/19/9/2785 ;https://www.hindawi.com/journals/omcl/2020/3836172/

The two articles that reviewer 2 suggests are review articles. Given it is a research article, we are unable to expand the health effects and the associated mechanism at that level. In the current manuscript, we cited our recent review article that discusses the broad range of health benefits and mechanisms of ellagic acid and its gut metabolite urolithins (reference #5). Also, there is one review article that discusses the health benefits of red raspberry in-depth (reference #4). These two review articles are cited in the Introduction section (lines 42-44, also see below).

Red raspberries have been studied extensively for their anti-inflammatory properties and their ability to attenuate inflammation and oxidative stress in multiple chronic diseases, including cardiovascular disease, obesity, cancer, and Alzheimer’s Disease [4,5].

  1. Burton-Freeman, B.M.; Sandhu, A.K.; Edirisinghe, I. Red Raspberries and Their Bioactive Polyphenols: Cardiometabolic and Neuronal Health Links. Adv Nutr 2016, 7, 44-65, doi:10.3945/an.115.009639.
  2. Kang, I.; Buckner, T.; Shay, N.F.; Gu, L.; Chung, S. Improvements in Metabolic Health with Consumption of Ellagic Acid and Subsequent Conversion into Urolithins: Evidence and Mechanisms. Adv Nutr 2016, 7, 961-972, doi:10.3945/an.116.012575.

  1. References:  Authors should revise the format of reference according to IJERPH Journal.

We have formatted the references according to IJERPH journal guidelines (MDP ACS downloaded EndNote style).

Reviewer 3 Report

This study investigated the neuroprotective effects of whole red raspberry polyphenols versus urolithin in BV-2 cells using two different time windows (3 hr versus 12~24 hr). The manuscript is well-founded, and the results are fascinating. However, there are a few issues that, if addressed, may further strengthen the manuscript.

  • The authors used acute and chronic terms. On what grounds was the term acute-chronic used when it calls 3 hours versus 12-24 hours.
  • According to the author’s description, BV-2 cells were pretreated with either RRW or UroA for 12 hours, and then LPL (+ATP) was treated for another 3 hours (acute) or another 12-24 hours (chronic). However, there was no supplementation of RRW or UroA during experiments. The RRW contains an array of bioactive components, including quercetin, myricetin, ellagic acid, and anthocyanins that are mostly hydrophobic with relatively short life half-life. For example, the average half-life of quercetin is only a few hours, and the ellagic acid half-life is even shorter than an hour. How about the half-life of UroA? The concentration changes of each and individual bioactive components in RRW during long-incubation time might cause reductions in bioactivities.
  • Some minor comments;
  • Lines 130-133, lines 155-157 are repeated information from the methods and introduction. Please remove or make it short if necessary.
  • The information on lines 138-140 would be better to move to the method section (line 93).
  • Correct an error in Figure 5 legend. (1) 24 hours ® 12 hours

Author Response

Reviewer 3

Comments and Suggestions for Authors

This study investigated the neuroprotective effects of whole red raspberry polyphenols versus urolithin in BV-2 cells using two different time windows (3 hr versus 12~24 hr). The manuscript is well-founded, and the results are fascinating. However, there are a few issues that, if addressed, may further strengthen the manuscript. 

  1. The authors used acute and chronic terms. On what grounds was the term acute-chronic used when it calls 3 hours versus 12-24 hours.

We thank the reviewer for their suggestion and have removed acute vs. chronic and replaced these terms with 3 hours versus 12-24 hours. As 3 hours reflects an ischemic model (acute) versus a chronic long-term inflammation such as Alzheimer’s Disease (12-24 hours), we had selected those terms. However, we have changed these terms to 3 hour and 12-24 hour to provide consistency.

  1. According to the author’s description, BV-2 cells were pretreated with either RRW or UroA for 12 hours, and then LPL (+ATP) was treated for another 3 hours (acute) or another 12-24 hours (chronic). However, there was no supplementation of RRW or UroA during experiments. The RRW contains an array of bioactive components, including quercetin, myricetin, ellagic acid, and anthocyanins that are mostly hydrophobic with relatively short life half-life. For example, the average half-life of quercetin is only a few hours, and the ellagic acid half-life is even shorter than an hour. How about the half-life of UroA? The concentration changes of each and individual bioactive components in RRW during long-incubation time might cause reductions in bioactivities.

We thank reviewer 3 for a considerate suggestion. Bioavailability is determined by multiple factors, including the absorption rate from the GI tract, transportation to the target tissue, conjugation with other molecules, and catabolism in the liver. In fact, UroA is formed slowly for several hours up to 24hrs after consumption of ellagic acid-containing food. It is assumed due to the variation in Uro-producing gut microbes in humans. Once it is formed, Uro seems to be stable.

We feel the bioavailability issue is difficult to be handled in vitro setting, and it might be out of scope since our study is exclusively in vitro setting. Also, our study is a very preliminary study to understand the role of red raspberry polyphenols in microglial cells. 

Some minor comments;

  1. Lines 130-133, lines 155-157 are repeated information from the methods and introduction. Please remove or make it short if necessary.

These changes have been updated (lines 134-135) and the sentence was shortened. We shortened lines 155-157 but want to emphasize urolithin A’s effect on JNK/c-Jun signaling pathways remained unknown.

  1. The information on lines 138-140 would be better to move to the method section (line 93).

Lines 138-140 was moved to the method section lines 94-97.

  1. Correct an error in Figure 5 legend. (1) 24 hours ® 12 hours

The error was corrected on line 237, Figure 5 legend.